

# Plagiarism detection across languages: a comprehensive study of Arabic and English-to-Arabic long documents

Ahmad Abdelaal[1], Abdallah Elsaadany[1], Abdelrhman Ahmed Medhat[1], Aysha Al Shamsi[2] and Noha Gamal ElDin Saad Ali[1]

[1] School of Information Technology and Computer Science, Nile University, Giza, Egypt
[2] Arabic Language Section, Mohamed Bin Zayed University for Humanities, Abu Dhabi, United Arab Emirates

## ABSTRACT

Plagiarism detection in Arabic texts remains a significant challenge due to the complex morphological structure, rich linguistic diversity, and scarcity of high-quality labeled datasets. This study proposes a robust framework for Arabic plagiarism detection by integrating Siamese neural networks (SNN) with state-of-the-art transformer architectures, specifically AraT5 and Longformer. The system employs a hybrid workflow, combining transformer-based encoders and a classification objective to implicitly learn textual similarity. To address the inherent imbalance in Arabic plagiarism datasets, both weighted cross-entropy loss and Dice loss functions were utilized to optimize model training. Extensive experiments were conducted using the ExAraCorpusPAN2015 dataset, demonstrating the effectiveness of the proposed architecture. Results indicate that AraT5 with weighted cross-entropy loss outperformed other configurations, achieving an F1-score of 0.9058. Additionally, comparative analysis with existing methodologies highlights the superiority of our approach in handling nuanced semantic and structural variations within Arabic texts. This study underscores the importance of transformer-based architectures and class-specific loss functions in enhancing plagiarism detection accuracy in under-resourced languages like Arabic.

## INTRODUCTION

Plagiarism detection has become an essential task in the digital era, driven by the rapid growth of accessible electronic content across academic, professional, and creative domains. While significant advancements have been made in plagiarism detection systems for languages such as English, the Arabic language remains vastly underexplored in this field (*Arabi & Akbari, 2022*; *Zouaoui & Rezeg, 2022*). This disparity not only undermines the academic integrity of Arabic content but also highlights a critical technological gap in natural language processing (NLP) tools tailored for Arabic text analysis.

The unique characteristics of Arabic—including its rich morphology, complex word inflections, and diverse dialects—pose significant challenges for automated plagiarism detection systems (*Nagoudi, Cherroun & Alshehri, 2018*). Arabic's script-based intricacies,

Corresponding authors
Ahmad Abdelaal,
a.abdelsalam@nu.edu.eg
Noha Gamal ElDin Saad Ali,
NGamal@nu.edu.eg

such as diacritics and varying orthographic styles, further complicate text alignment and semantic analysis tasks (*Alzahrani & Aljuaid, 2022*). Moreover, datasets curated specifically for Arabic plagiarism detection are limited, often exhibiting imbalanced distributions and a lack of standardized annotation (*Bensalem et al., 2015*; *Al-Sulaiti & Atwell, 2006*).

Benchmark datasets and shared tasks have significantly advanced plagiarism detection research. The AraPlagDet shared task (*Bensalem et al., 2015*) provides a platform for evaluating Arabic plagiarism detection systems, promoting comparative research and system development. Complementary resources, such as the Contemporary Arabic Corpus (CCA) (*Al-Sulaiti & Atwell, 2006*), offer linguistic datasets for training and evaluating these systems, facilitating progress in the field. However, these resources remain insufficient to fully address the growing demand for robust Arabic plagiarism detection systems.

Existing systems frequently rely on traditional text-matching techniques, statistical methods, or machine translation layers for cross-lingual plagiarism detection (*Alotaibi & Joy, 2021*). However, these approaches often fail to address deeper semantic obfuscations, paraphrasing complexities, and scalability for long-document analysis. Despite the advent of transformer-based architectures, such as Longformer (*Beltagy, Peters & Cohan, 2020*) and Arabic-specific models like AraT5 (*Elmadany, Nagoudi & Abdul-Mageed, 2023*), their application to Arabic plagiarism detection remains underexplored, particularly in cross-lingual scenarios.

This study aims to bridge this technological and methodological gap by addressing the key challenges of Arabic plagiarism detection. Our work emphasizes the importance of handling long-document sequences, overcoming dataset imbalances, and improving semantic similarity analysis in Arabic and cross-lingual contexts. A detailed overview of our contributions, including specific advancements and methodologies, is provided in 'Gaps in related work and proposed contributions'.

The rest of this article is organized as follows: 'Related Work' reviews existing literature on plagiarism detection and related NLP advancements. 'Dataset' introduces the dataset and preprocessing strategies. 'Methodology' details the methodology and model architectures. 'Results and Discussion' presents the experimental setup, evaluation results, and comparative analysis. Finally, 'Conclusion' concludes with key findings, limitations, and future research directions.

## RELATED WORK

Text similarity analysis underpins critical applications in plagiarism detection, machine translation evaluation, and text generation assessment. This section discusses advancements in plagiarism detection and neural architectures for similarity analysis, alongside evaluation metrics, to provide a comprehensive overview of the field.

### Plagiarism detection

Plagiarism detection systems aim to identify similarities between textual content, which may occur within the same language (monolingual plagiarism) or across different languages (cross-lingual plagiarism). These systems must overcome challenges such as

linguistic diversity, disguised textual modifications, and scalability. The detection of cross-language plagiarism, especially in the context of machine translation and multilingual digital content, is becoming increasingly critical. Traditional monolingual methods struggle with sophisticated forms of plagiarism involving translation. While some methods attempt to use machine translation to unify texts into a common language, they are limited by translation inaccuracies. Recent approaches, therefore, focus on semantic and syntactic analysis, albeit at higher computational costs and requiring parallel corpora.

## Monolingual plagiarism detection

Monolingual plagiarism detection traditionally relied on exact and approximate text matching techniques. However, recent research emphasizes hybrid approaches that integrate multiple similarity measures to improve robustness and accuracy.

*Arabi & Akbari (2022)* propose two approaches for monolingual plagiarism detection. The first combines FastText pre-trained word embeddings with term frequency-inverse document frequency (TF-IDF) weighting to form structural-semantic matrices of sentences. The second employs WordNet ontology alongside TF-IDF weighting to create similar matrices. Both approaches use a two-stage filtering process at the document and sentence levels, leveraging the Bag of Words (BoW) technique to narrow the search space. After filtering, structural and semantic matrices are formed, and their similarity is calculated using the Dice similarity measure. Documents exceeding a predefined similarity threshold are flagged as plagiarized. Validated on the PAN-PC-11 dataset, their FastText-based method achieved an F1-score of 90.97, while the WordNet-based method reached an F1-score of 90.00. These results highlight the effectiveness of combining lexical and semantic features for detecting sophisticated forms of plagiarism.

Another innovative approach is the multi-agent indexing system (MAIS) introduced by *Zouaoui & Rezeg (2022)*. This index-based framework integrates semantic indexing with multi-agent systems, utilizing the Sorensen-Dice similarity measure for document retrieval. MAIS leverages Arabic ontology (ShemNet) to establish semantic closeness between entities in documents. The system proved effective at identifying complex types of plagiarism, such as synonym replacement and sentence scrambling, achieving superior performance on the AraPlagDet dataset.

*Nagoudi, Cherroun & Alshehri (2018)* investigated disguised plagiarism, focusing on advanced obfuscation techniques such as paraphrasing, word replacement, and structural manipulation in Arabic texts. Their research employs word embeddings and machine learning models, including support vector machines (SVM) and Random Forests, alongside sophisticated NLP techniques like bag-of-meanings and semantic similarity measures. The combination of lexical, syntactic, and semantic features proved effective in identifying complex plagiarism patterns.

*Aliane & Aliane (2020)* investigated the effectiveness of Siamese neural architectures for Arabic semantic textual similarity (STS) and plagiarism detection. Their study evaluated three Siamese-based models: bidirectional long short-term memory network (BiLSTM), convolutional neural network (CNN), and a transformer-based model using AraBERT. Each model consisted of a shared architecture for encoding sentence pairs and a

comparison layer using similarity metrics like cosine similarity and dot product. They tested their models on two Arabic datasets: the PAN 2015 *corpus* for plagiarism detection and the Mawdoo3 question similarity dataset. The results demonstrated that the transformer model consistently outperformed the others, achieving a Pearson correlation of up to 0.93 on the PAN dataset.

Their work confirms the relevance of Siamese networks for Arabic STS and highlights the superiority of contextualised transformer-based embeddings in low-resource settings. While their focus was on sentence and short paragraph comparison, their findings provide a foundational baseline for extending Siamese architectures to longer document-level tasks, such as plagiarism detection. By demonstrating the robustness of AraBERT in the Siamese setting, this work paves the way for exploring more scalable transformer variants like Longformer to handle longer Arabic texts more effectively.

*Saidi, Jarray & Alsuhaibani (2023)* proposed the Siamese bidirectional encoder representations from transformers (SiameseBERT) a BERT-based Siamese network enhanced with a soft attention mechanism to assess semantic textual similarity (STS) for Arabic. Their approach embeds input sentences using various Arabic BERT models, including AraBERT and CAMeL-BERT, and applies a soft alignment mechanism to capture word-level interactions between sentence pairs. The model architecture integrates contextual embeddings through a Siamese structure, followed by attention-driven interaction, pooling, and prediction layers. Their experiments on benchmark Arabic STS datasets demonstrated the effectiveness of this architecture, achieving a high Pearson correlation score of up to 0.925 on the MSR-Paraphrase dataset.

The study highlights the strength of combining BERT embeddings, attention mechanisms, and Siamese networks for Arabic STS—a task crucial for applications such as plagiarism detection. It also shows that the inclusion of an attention mechanism consistently improves model performance across datasets. While their work focused on sentence-level similarity using transformer-based models limited by input length, it provides a strong foundation for exploring architectures like Longformer that address longer textual inputs, which are essential in document-level tasks such as Arabic plagiarism detection.

## Cross-lingual plagiarism detection

Cross-lingual plagiarism detection refers to the identification of semantically equivalent or paraphrased content that has been translated between different languages. Unlike monolingual plagiarism, which compares text in the same language, cross-lingual detection must account for variations introduced by translation, including differences in syntax, morphology, and idiomatic expressions. This task typically involves a preprocessing step such as machine translation to bring texts into a common representation space, followed by semantic similarity assessment.

In the English-Arabic context, this challenge is further amplified due to the rich morphology of Arabic and structural asymmetry between the two languages. Therefore, effective cross-lingual detection systems must either incorporate a robust translation layer or utilize multilingual encoders capable of capturing aligned semantics across languages.

The following studies illustrate some of the techniques used in this emerging area of research. *Alzahrani & Aljuaid (2022)* address these challenges by employing deep neural networks equipped with semantic-rich features, such as topic similarity, named entity recognition (NER), semantic role labeling (SRL), and bag-of-meanings (BoM). Their models, configured with two or more hidden layers and Rectified Linear Unit (ReLU) activation functions, are trained for binary classification tasks using binary cross-entropy loss. Additionally, they implement multi-class classification for translation types (*e.g.*, literal, paraphrased, summarized, independently written) using categorical cross-entropy. Their experiments demonstrated that all deep neural network models outperformed traditional classifiers like SVM or logistic regression, achieving a maximum classification accuracy of 97.01%. For cross-lingual semantic similarity prediction, they obtained an accuracy of 71.13%.

*Alotaibi & Joy (2021)* contribute to English-Arabic plagiarism detection by extracting semantic and syntactic features integrated into machine learning classifiers with multilingual encoders. Their system evaluates these features on the SemEval-2017 dataset, revealing that the support vector classifier (SVC) yielded the best F1-score of 0.879. This study underscores the potential of integrating linguistic features with robust classification models to address the structural divergence between English and Arabic.

Additional contributions include the work by *Aljuaid (2020)*, who proposed a word-embedding and IDF-weighted model for English–Arabic plagiarism detection using comparable corpora, achieving over 82% sentence-level accuracy. Earlier, *Aljohani & Mohd (2014)* introduced an English-Arabic cross-language detection method using machine translation followed by a winnowing fingerprinting algorithm, reporting a recall of 81% and precision of 97%. These studies illustrate the diverse techniques—spanning embedding models, translation layers, and fingerprinting—that have been employed to address translation-induced variations in English–Arabic documents.

Advances in neural architectures have revolutionized text similarity analysis, enabling models to capture complex semantic relationships. For example, Unified transformer models like T5 (*Raffel et al., 2020*) simplify NLP by framing all tasks as text-to-text problems. *Elmadany, Nagoudi & Abdul-Mageed (2023)* extend this paradigm with Octopus, a multitask model tailored for Arabic NLP tasks. By leveraging task-specific fine-tuning, Octopus enhances performance on low-resource languages, making it a promising tool for future similarity analysis and text generation. The studies reviewed here illustrate a dynamic and evolving field, addressing critical challenges in monolingual and cross-lingual plagiarism detection, metric development, and neural model design. Despite significant progress, open challenges include scalability, computational efficiency, and the detection of obfuscated plagiarism, which remain areas for further exploration.

## Gaps in related work and proposed contributions

The proposed research builds upon the foundations laid by previous studies while addressing significant gaps in the field of Arabic plagiarism detection. Table 1 provides a comprehensive comparison between the surveyed studies and the proposed work, focusing

**Table 1 Gaps of surveyed research *vs.* proposed work contributions.**

| Aspect | Gaps of existing research | Contributions of proposed work |
|---|---|---|
| Dataset | – Limited label diversity and imbalanced distributions.<br>– Restricted token limits in traditional models. | – Augments ExAraCorpusPAN2015 dataset with a "No Plagiarism" label.<br>– Balances dataset distribution during preprocessing.<br>– Adjusts context size to fit transformer limits (1,024 for AraT5 and 4,096 for Longformer). |
| Methodology | – Employs traditional techniques like BoW and word embeddings (*Arabi & Akbari, 2022*; *Nagoudi, Cherroun & Alshehri, 2018*).<br>– Ontology-based systems like MAIS (*Zouaoui & Rezeg, 2022*).<br>– Multilingual encoders for cross-lingual tasks (*Alotaibi & Joy, 2021*). | – Leverages transformer architectures (Longformer and AraT5) for improved contextual understanding.<br>– Siamese neural networks for semantic similarity between document pairs.<br>– Incorporates a robust translation layer for cross-lingual capabilities. |
| Complex plagiarism types | – Focused on specific types (*e.g.*, synonym replacement, paraphrasing) but lacks integration of multiple types (*Nagoudi, Cherroun & Alshehri, 2018*; *Zouaoui & Rezeg, 2022*). | – Detects various types (No Obfuscation, Artificial Obfuscation, Simulated Obfuscation) within a unified framework.<br>– Tailors detection methods to Arabic linguistic nuances and morphological complexity. |
| Scalability | – Limited by computational inefficiencies in large documents (*Arabi & Akbari, 2022*).<br>– Traditional models struggle with long-context documents. | – Longformer handles up to 4,096 tokens, making it well-suited for long-document plagiarism detection.<br>– Efficient use of sliding-window attention for scalability. |
| Cross-lingual plagiarism | – Uses machine translation with inherent inaccuracies (*Alotaibi & Joy, 2021*).<br>– Semantic features combined with deep learning (*Alzahrani & Aljuaid, 2022*). | – Introduces a rigorously evaluated translation layer, selecting Argos-Translate for its high performance.<br>– Ensures effective integration of multilingual data into Arabic-focused detection models. |

on dataset usage, methodology, ability to handle complex plagiarism types, scalability, cross-lingual capabilities, and evaluation rigor.

# DATASET

The Evaluation *Corpus* for Arabic External Plagiarism Detection (ExAraCorpusPAN2015) was introduced as part of the AraPlagDet shared task (*Bensalem et al., 2015*), it was the only publicly available dataset found at the time of working on this research, which is available at https://doi.org/10.5281/zenodo.6607799. This dataset was meticulously constructed using two primary sources: documents from Wikipedia and the CCA (*Al-Sulaiti & Atwell, 2006*). The CCA is a well-curated linguistic resource aimed at providing a large-scale, freely accessible *corpus* of modern Arabic texts, spanning various domains and writing styles. The ExAraCorpusPAN2015 dataset consists of 1,725 documents explicitly annotated for external plagiarism detection. Each document in the dataset is labeled according to one of three primary plagiarism types:

- **No obfuscation:** The plagiarized text appears in its original form with no significant modifications.

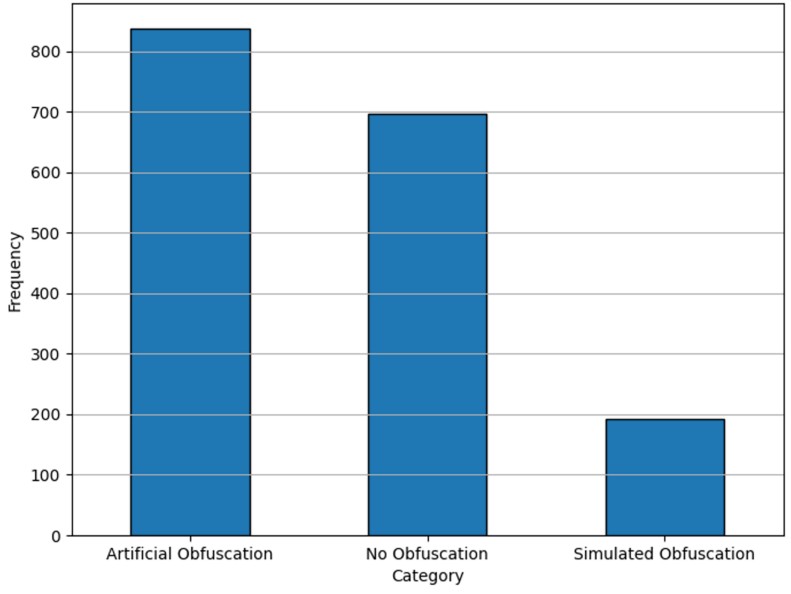

**Figure 1** **Label distributions for raw data.**

- **Artificial obfuscation:** The plagiarized text undergoes deliberate lexical and structural alterations to disguise its origin.
- **Simulated obfuscation:** The plagiarized text features paraphrasing, sentence restructuring, and other sophisticated transformations to obscure plagiarism traces.

Despite its valuable contributions, the dataset exhibits inherent challenges, including:

1. **Class imbalance:** A significant skew in label distribution, with certain classes dominating the dataset.
2. **Token variability:** Wide variation in document lengths, with some exceeding typical model context limits.

This imbalance issue is evident in the initial label distribution, as illustrated in Fig. 1. Addressing these challenges was a crucial step in our preprocessing pipeline, which is discussed in detail in 'Data pre-processing & feature engineering'. The ExAraCorpusPAN2015 dataset serves as a foundational resource for benchmarking plagiarism detection systems in Arabic, offering a diverse representation of plagiarism types and complexities. It provides a robust testing ground for our proposed system, ensuring its applicability to real-world scenarios and varied plagiarism patterns.

## METHODOLOGY

This section outlines the methodological approach adopted for developing our plagiarism detection system. The workflow is structured into distinct stages: preprocessing the dataset, implementing a translation layer for cross-lingual compatibility, designing the Siamese

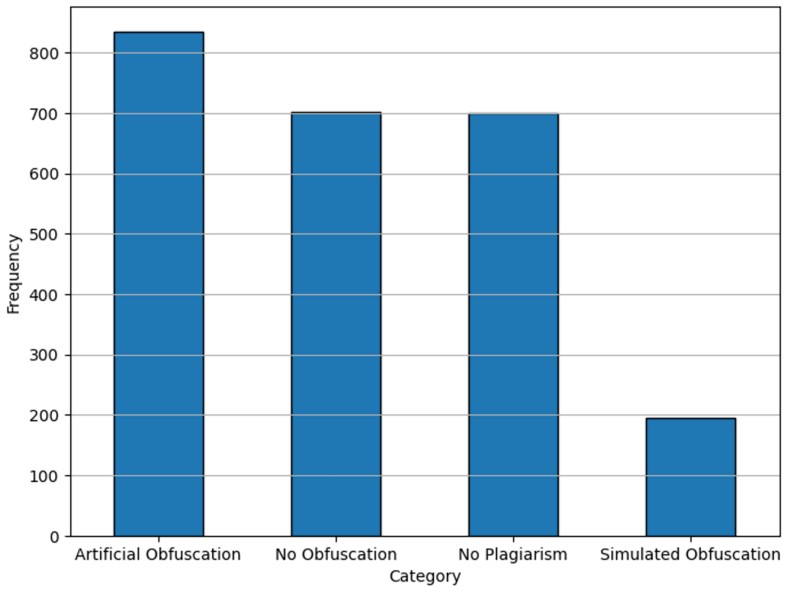

**Figure 2** **Plagiarism detection workflow.**

architecture using Longformer and AraT5 models, and evaluating system performance through appropriate metrics.

## Plagiarism detection workflow

The proposed workflow aims to detect potential plagiarism across Arabic documents and cross-lingual sources. The system operates in three main stages:

1. **Language standardization:** Input text pairs (source and suspicious documents) are first analyzed to ensure they are both in Arabic. If one document is non-Arabic, a translation layer converts it into Arabic to maintain consistency.

2. **Feature representation:** The Arabic text pairs are passed through a transformer-based Siamese network architecture (using either Longformer or AraT5 encoders). These models are fine-tuned specifically for Arabic text to capture both semantic and structural relationships.

3. **Classification:** The embeddings generated by the Siamese network are compared and passed through a classification head to determine whether plagiarism exists and, if so, to classify the type of plagiarism (*e.g.*, No Obfuscation, Artificial Obfuscation, Simulated Obfuscation).

The overall workflow is illustrated in Fig. 2, depicting the sequential stages from input document processing to final plagiarism classification.

This modular pipeline ensures robust handling of linguistic variations, cross-lingual text pairs, and complex plagiarism patterns, providing a flexible and scalable foundation for document similarity analysis.

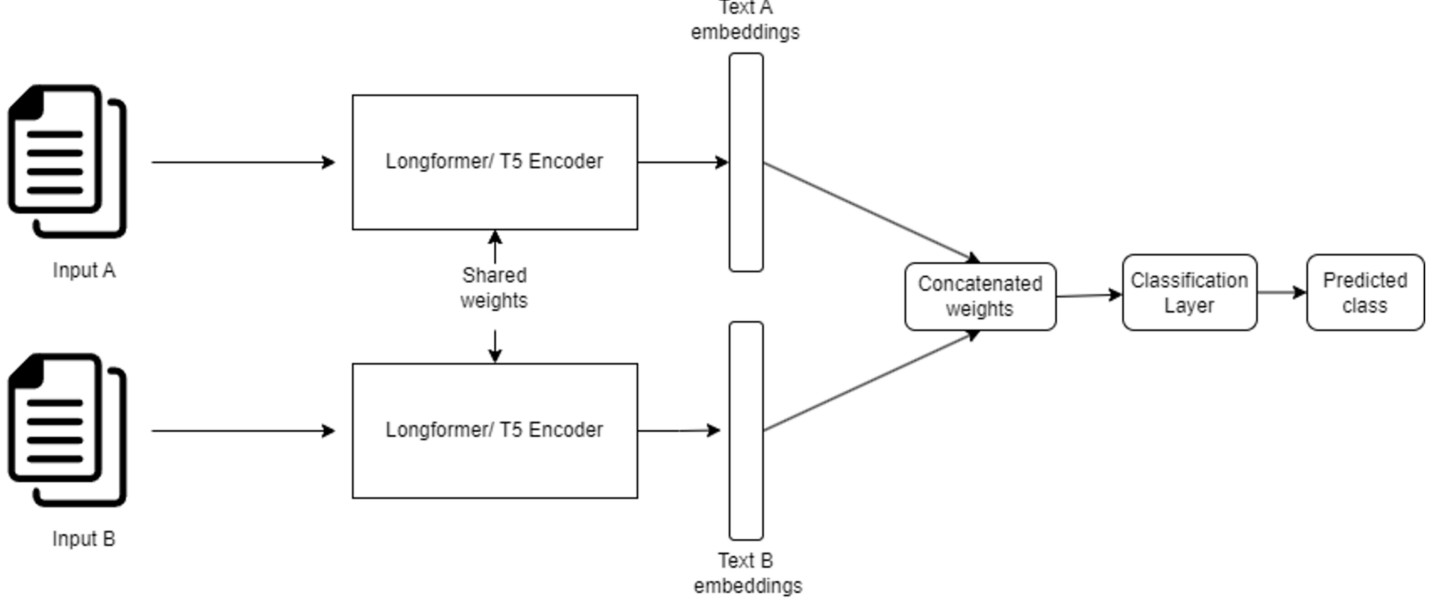

**Figure 3 Label distributions for dataset after preprocessing.**

### Data pre-processing and feature engineering

Preprocessing is a crucial step to address dataset imbalances and adapt text inputs for the limitations of transformer-based architectures. The preprocessing phase involves the following key steps:

1. **Context size management:** Due to the limited context window of Longformer (4,096 tokens) and AraT5 (1,024 tokens), each document was truncated to fit within these token limits. Careful attention was given to ensure that plagiarized text offsets remain within these truncated contexts to prevent loss of critical information.

2. **No-plagiarism label creation:** A "No Plagiarism" label was introduced by extracting non-overlapping text segments from the dataset where no plagiarism indicators were present. This additional label enhances the model's ability to differentiate between genuine and plagiarized content.

3. **Tokenization and encoding:** Text pairs were tokenized using their respective model tokenizers (LongformerTokenizer or T5Tokenizer) to ensure compatibility with the downstream architecture.

The effects of these preprocessing steps are evident in the adjusted label distributions shown in Fig. 3. These steps collectively ensure that the dataset is optimized for transformer-based architectures, mitigating common pitfalls such as sequence truncation and class imbalance.

### Translation layer

The integration of a translation layer in our plagiarism detection system addresses the challenge of cross-lingual plagiarism, particularly when source and suspicious documents

are written in different languages. This layer ensures linguistic uniformity by converting non-Arabic text into Arabic, allowing seamless processing by Arabic-specific NLP models. Given the structural and syntactic complexities of the Arabic language, selecting an optimal translation system plays a pivotal role in preserving both semantic and contextual fidelity during translation.

In this study, we evaluated multiple state-of-the-art translation models, including both commercial APIs and open-source large language models (LLMs), based on their suitability for the Arabic language. These models represent different architectures, ranging from sequence-to-sequence neural networks to transformer-based systems, and exhibit varying strengths in handling context-rich translations, idiomatic expressions, and morphological variations.

Commercial translation systems, such as Google Translate, are widely recognized for their robust multilingual capabilities and are built upon extensive neural machine translation (NMT) frameworks (*Wu et al., 2016*). However, their performance often varies across languages, especially for morphologically rich languages like Arabic. Similarly, Argos Translate, an open-source translation toolkit, leverages pre-trained neural network models optimized for multilingual tasks and provides customization capabilities for domain-specific translations (*Finlay & Argos Translate Contributors, 2025*).

In contrast, recent advancements in transformer-based models, such as mBART-Large-50-Many-to-Many-MMT, have shown promise in addressing cross-lingual tasks by extending pre-training to multiple languages (*Liu et al., 2020*). mBART employs a denoising autoencoder mechanism for sequence-to-sequence tasks, which improves its ability to maintain syntactic and semantic fidelity during translation. Another significant contribution comes from models like Gemini Flash 1.5 and SILMA-9B-Instruct-v1.0, both fine-tuned on multilingual corpora and optimized for translation-specific tasks. Gemini Flash 1.5, a generative transformer-based language model, incorporates extensive contextual embeddings to address nuances specific to Arabic, while SILMA-9B-Instruct-v1.0 adapts its decoder architecture for multilingual translation scenarios (*Qian et al., 2024*).

Additionally, Marefa-MT-En-Ar, a specialized English-to-Arabic translation model, was included in our evaluation due to its focus on Arabic's unique linguistic features. Marefa-MT utilizes domain-specific vocabulary and fine-tuned attention mechanisms to ensure that translations retain grammatical coherence and stylistic appropriateness in Arabic text (*marefa-nlp, 2023*).

The translation layer, therefore, serves as a critical intermediary component, ensuring that non-Arabic content can be seamlessly integrated into our Arabic-focused plagiarism detection pipeline. By systematically evaluating multiple translation models, we have ensured that the selected system not only meets linguistic accuracy requirements but also minimizes semantic drift, thereby enhancing the reliability of the overall detection process.

### Translation layer model evaluation

To select the most suitable model for our translation layer, we employed the OpenSubtitles dataset by *Lison & Tiedemann (2016)*, which contains parallel corpora in multiple

languages. Specifically, we used the English–Arabic sentence pairs and randomly sampled 10% of the data to perform inference across all the previously discussed models. This evaluation allowed us to assess which model produced the most syntactically and semantically accurate translations from the following evaluation metrics, serving as a basis for selecting the optimal translation system for our pipeline.

To objectively evaluate these translation systems, we employed established evaluation metrics, including BLEU (Bilingual Evaluation Understudy) (*Papineni et al., 2002*), chrF (Character-Level F-score) (*Popović, 2015*), BERTScore (*Zhang et al., 2020*), and METEOR (Metric for Evaluation of Translation with Explicit Ordering) (*Banerjee & Lavie, 2005*). These metrics were chosen for their ability to measure different aspects of translation quality, such as lexical overlap, semantic similarity, and syntactic alignment as will be detailed in the next subsection.

The BLEU (Bilingual Evaluation Understudy) metric (*Papineni et al., 2002*) was utilized to measure the overlap of n-grams between the translated text and reference text. BLEU operates at the word and phrase level, offering an effective measure of lexical similarity, but it may struggle to capture deeper semantic relationships.

To address lexical limitations, the chrF (Character-Level F-score) metric (*Popović, 2015*) was included in the evaluation. chrF calculates F-scores based on character n-grams rather than words, making it more suitable for morphologically rich languages such as Arabic, where word forms vary significantly.

Further semantic evaluation was performed using BERTScore-F1 (*Zhang et al., 2020*). Unlike BLEU and chrF, which focus primarily on surface-level similarity, BERTScore utilizes contextual embeddings from pre-trained transformer models to compare sentence-level semantics. This enables a more nuanced understanding of translation quality by capturing relationships between words in different contexts.

Lastly, the METEOR (Metric for Evaluation of Translation with Explicit ORdering) metric (*Banerjee & Lavie, 2005*) was applied to complement the previous evaluations. METEOR considers not only n-gram overlap but also synonymy, stemming, and alignment of chunks between reference and translated texts. This multi-faceted approach ensures a balanced assessment of both lexical and semantic aspects of translation.

By combining these four evaluation metrics, the translation systems were assessed for both syntactic fidelity and semantic accuracy, providing a robust framework for identifying the most suitable translation model for the plagiarism detection pipeline.

## Plagiarism detection layer

In this section, we present the core architecture utilized for detecting plagiarism across Arabic documents. At the heart of our system lies a Siamese neural network (SNN) (*Bromley et al., 1993*), a well-established framework for similarity tasks, adapted here to address the nuances of Arabic textual data.

### Siamese network

SNNs have emerged as a powerful architecture for measuring the similarity between two inputs. Originally proposed for tasks such as signature verification (*Bromley et al., 1993*),

they have been effectively adapted for applications in text similarity and plagiarism detection. In our implementation, a Siamese network is employed to compare two text documents and classify their relationship into predefined categories, such as "No Plagiarism," "Artificial Obfuscation," and "Simulated Obfuscation."

**Architecture overview** The SNN consists of two identical sub-networks that share the same architecture, weights, and parameters. Each sub-network processes one of the input documents independently, transforming it into a fixed-size embedding vector. These embeddings are then concatenated and passed through fully connected dense layers for classification.

The Siamese neural network architecture employed in this study offers several key advantages for plagiarism detection tasks. By utilizing shared weights across both input branches, the network ensures symmetric processing of document pairs, reducing computational overhead and guaranteeing consistent feature extraction. Additionally, the architecture excels in capturing semantic similarity, making it highly effective in identifying subtle differences between documents, such as paraphrased text or structural variations.

The integration of Transformer-based encoders (AraT5 and Longformer) enhances the network's capability to process both monolingual and cross-lingual text pairs. The embeddings generated by these encoders play a pivotal role in capturing intricate structural and semantic relationships. Furthermore, the adoption of weighted cross-entropy loss and Dice loss effectively mitigates class imbalance issues, improving the network's predictive accuracy across different types of plagiarism.

This carefully designed architecture, illustrated in Fig. 4, provides a robust foundation for accurately detecting and categorizing plagiarism in Arabic and cross-lingual text datasets.

**1. Document encoding:** The two input documents, denoted as $X_1$ and $X_2$, are passed through a shared transformer-based encoder, such as Longformer (*Beltagy, Peters & Cohan, 2020*) or AraT5 (*Elmadany, Nagoudi & Abdul-Mageed, 2023*). The encoder produces contextual embeddings for each document:

$$E_1 = f_\theta(X_1), \quad E_2 = f_\theta(X_2) \tag{1}$$

where:
– $f_\theta$ represents the shared transformer encoder, and
– $E_1$, $E_2$ are the contextual embeddings of the input documents.

**2. Mean pooling:** To produce a fixed-size representation for each document, mean pooling is applied across the token embeddings:

$$E_1 = \frac{1}{n} \sum_{i=1}^{n} A_{1i}, \quad E_2 = \frac{1}{m} \sum_{j=1}^{m} A_{2j} \tag{2}$$

where:
– $n$ and $m$ are the token lengths of the documents.

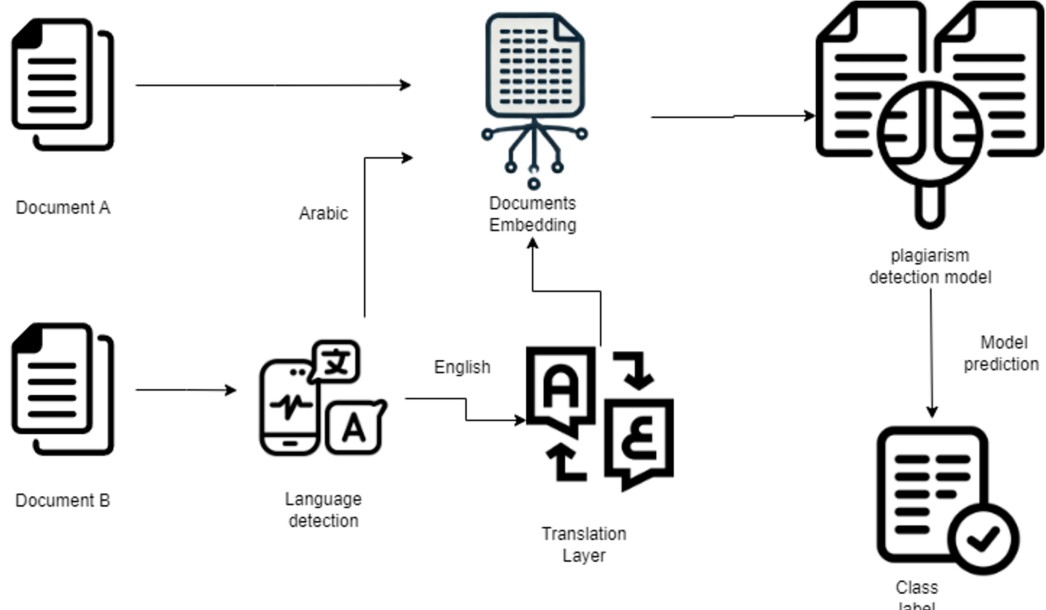

**Figure 4 Siamese neural network architecture for plagiarism detection.**

– $A_{1i}$ and $A_{2j}$ represent the embeddings of individual tokens in $X_1$ and $X_2$, respectively.
The mean pooling operation ensures that variable-length input sequences are mapped to fixed-size vector embeddings.

**3. Embedding concatenation:** The resulting document embeddings, $E_1$ and $E_2$, are concatenated into a single vector:

$$E = [E_1 \| E_2] \tag{3}$$

This concatenated vector $E$ serves as the combined representation of the document pair.

**4. Dense layers and classification:** The concatenated embeddings are passed through a series of dense layers, where transformations are applied to extract high-level semantic relationships:

$$H = \text{ReLU}(W_1 E + b_1) \tag{4}$$
$$H' = \text{Dropout}(H) \tag{5}$$
$$y = \text{Softmax}(W_2 H' + b_2) \tag{6}$$

where:
– $W_1$ and $W_2$ are weight matrices for the dense layers.
– $b_1$ and $b_2$ are bias terms.

– $H$ represents the hidden layer activations after applying the ReLU non-linearity.

– $H'$ represents the output after dropout regularization, and
– $y$ is the output probability distribution over the predefined classes.

**5. Similarity representation:** In our implementation, similarity between document embeddings is primarily learned implicitly through the dense layers, which are optimized for a classification objective rather than explicit similarity computation. The dense layers are trained to map the embeddings into a space where relationships between document pairs correspond to their similarity class labels (*e.g.*, *No Plagiarism*, *Artificial Obfuscation*, *Simulated Obfuscation*). Our approach relies on the final softmax classification layer to determine similarity categories. This layer inherently learns a more complex representation of semantic and structural similarities during the training process, guided by the chosen loss functions (*weighted cross-entropy* or *Dice loss*).

**6. Loss function:** The network is optimized using either weighted cross-entropy loss or Dice loss, both of which address the class imbalance issue in our dataset as will be detailed in section "Loss functions".

**7. Prediction:** The final predicted class is determined by selecting the maximum probability from the output layer:

$$\hat{y} = \arg\max(y) \tag{7}$$

This Siamese network, combined with transformer-based encoders and an effective loss function, provides a robust foundation for accurately detecting and classifying plagiarism types across Arabic and cross-lingual datasets.

### Language models

Transformer architectures have revolutionized NLP, enabling state-of-the-art performance across a wide range of text-related tasks. However, traditional transformer models, such as BERT (*Devlin, Chang & Toutanova, 2019*), suffer from significant limitations when processing long documents due to their quadratic complexity in memory and computational requirements, scaling with the square of the input sequence length. These limitations pose challenges in applications like document-level plagiarism detection, where capturing long-range dependencies and global contextual relationships is crucial.

**Longformer: efficient long-context representation:** The Longformer (*Beltagy, Peters & Cohan, 2020*) was introduced to address these limitations by replacing the traditional self-attention mechanism with a more efficient *sparse attention* mechanism. This innovation enables the Longformer to scale linearly with sequence length, making it feasible to process significantly longer documents without exhausting memory resources. The model utilizes two key attention strategies:

- **Sliding-window attention:** Each token attends to a fixed-size window of surrounding tokens, capturing local dependencies efficiently.
- **Global attention:** Selected tokens (*e.g.*, classification tokens) attend to all other tokens, allowing the model to capture global dependencies critical for tasks requiring document-level understanding.

Formally, given an input sequence $X = [x_1, x_2, \ldots, x_n]$, the Longformer computes the representation of each token as:

$$h_i = \text{Attention}(x_i, \mathscr{W}(x_i), \mathscr{G}(x_i)) \qquad (8)$$

where:

– $x_i$ is the embedding of token $i$.
– $\mathscr{W}(x_i)$ represents the local sliding window attention over neighboring tokens.
– $\mathscr{G}(x_i)$ represents global attention for designated tokens.

This combination allows the Longformer to model both fine-grained local details and long-range dependencies efficiently, making it particularly well-suited for plagiarism detection tasks that require semantic similarity comparison across extended text spans. For our implementation, the Longformer processes document pairs with a maximum sequence length of 4,096 tokens, ensuring that essential context is retained during analysis.

**AraT5: Tailored for Arabic NLP** The AraT5 family of models, including the latest AraT5 (*Elmadany, Nagoudi & Abdul-Mageed, 2023*), is specifically designed to address the linguistic intricacies of the Arabic language. Built upon the Text-to-Text Transfer Transformer (T5) architecture (*Raffel et al., 2020*), AraT5 reframes every NLP task as a text-to-text problem, enabling a unified framework for classification, translation, and summarization.

The AraT5 model offers key enhancements over its predecessor:

- **Increased context size:** Supports input sequences up to 1,024 tokens, effectively doubling the previous limit of 512 tokens.
- **Arabic-specific pretraining:** Pretrained on a diverse *corpus* of Arabic text, covering Modern Standard Arabic (MSA), colloquial dialects, and domain-specific datasets.
- **Improved linguistic representation:** Fine-tuned on Arabic linguistic tasks, enabling superior performance in capturing syntactic and semantic relationships unique to the Arabic language.

Given the limited availability of transformer models explicitly designed for Arabic, AraT5 provides a robust foundation for capturing the nuances of Arabic text. In our implementation, AraT5 is employed in scenarios where shorter text spans (up to 1,024 tokens) require deep contextual analysis.

**Comparison and integration** While the Longformer excels in processing long documents with extensive context windows, AraT5 offers a refined linguistic representation tailored to the Arabic language. The choice between these models depends on the context length and linguistic complexity of the input texts. The embeddings generated by these models are further utilized within our Siamese network architecture for downstream similarity analysis and classification tasks.

The integration of both models in our framework ensures flexibility and robustness:

$$E_1 = f_{\text{encoder}}(X_1), \quad E_2 = f_{\text{encoder}}(X_2) \qquad (9)$$

where $f_{\text{encoder}}$ represents either the Longformer or AraT5 encoder. The resulting embeddings are subsequently processed by the Siamese network to identify and classify plagiarism patterns effectively.

By leveraging these two complementary transformer models, our system achieves a balanced trade-off between contextual richness (Longformer) and linguistic accuracy (AraT5), addressing both scalability and language-specific challenges inherent to Arabic plagiarism detection.

### Loss functions

Handling imbalanced labels in the dataset is a critical challenge in achieving effective model training and ensuring accurate predictions. To address this issue, we trained our model twice, using two different loss functions: Weighted Cross Entropy Loss and Dice loss. These loss functions were chosen for their ability to mitigate the impact of class imbalance and enhance the model's ability to generalize across all labels.

**Weighted cross entropy loss:** Weighted Cross Entropy Loss extends the standard Cross Entropy Loss by introducing class-specific weights to address imbalanced datasets. In this approach, weights are assigned to each class based on their frequency in the dataset, with higher weights given to underrepresented classes. This modification ensures that minority classes contribute more significantly to the loss function, thereby reducing the bias toward majority classes during training.

$$\mathscr{L}_{CE} = -\sum_{i=1}^{C} w_i y_i \log(\hat{y}_i) \tag{10}$$

where:
– $y_i$ and $\hat{y}_i$ represent the true and predicted class probabilities.

– $C$ is the number of classes.
– $w_i$ represents class weights.

**Dice loss:** Dice loss is specifically designed to handle imbalanced data by focusing on the overlap between predicted and true labels. Based on the Dice coefficient, it measures the ratio of intersection to the total size of the predicted and ground truth sets, making it a direct optimization for balance between precision and recall. Unlike weighted cross entropy loss, Dice loss inherently gives equal importance to all classes without requiring explicit weighting.

$$\mathscr{L}_{Dice} = 1 - \frac{2\sum(p \cdot t) + \varepsilon}{\sum(p + t) + \varepsilon} \tag{11}$$

where:
– $p$ represents the predicted probabilities.
– $t$ represents the true labels.
– $\varepsilon$ is a smoothing factor to prevent division by zero.

## Evaluation metrics

**Precision:** Precision evaluates the proportion of correctly predicted positive instances out of all predicted positive instances. It measures the model's accuracy in making positive predictions and is defined as:

$$\text{Precision} = \frac{\text{TP}}{\text{TP} + \text{FP}} \tag{12}$$

where *TP* is the number of true positives (correctly predicted positive instances), and *FP* is the number of false positives (incorrectly predicted positive instances).

**Recall:** Recall (or Sensitivity) measures the proportion of correctly predicted positive instances out of all actual positive instances. It reflects the model's ability to identify all relevant instances and is defined as:

$$\text{Recall} = \frac{\text{TP}}{\text{TP} + \text{FN}} \tag{13}$$

where *TP* is the number of true positives, and *FN* is the number of false negatives (actual positive instances incorrectly predicted as negative).

**F1-score:** F1-score is the harmonic mean of precision and recall, providing a balanced measure of a model's accuracy when dealing with imbalanced datasets. It is particularly useful when the dataset has a skewed distribution of classes. F1-score is defined as:

$$\text{F1-score} = 2 \cdot \frac{\text{Precision} \cdot \text{Recall}}{\text{Precision} + \text{Recall}} \tag{14}$$

These metrics were chosen to comprehensively evaluate the models, as they provide insights into different aspects of performance. Precision emphasizes minimizing false positives, Recall focuses on reducing false negatives, and F1-score offers a balanced evaluation, especially when dealing with class imbalances in the dataset.

## Hardware and software infrastructure

All experiments were conducted on a system equipped with an AMD Ryzen 9 5900X processor, 128 GB of RAM, and an NVIDIA RTX 3090 GPU. The experiments were implemented using Python version 3.11.0. To maintain a consistent environment for all experiments, the same hardware and software configurations were used throughout the study.

## RESULTS AND DISCUSSION

### Translation layer results

The performance of each translation model was evaluated using the previously mentioned metrics, and the results are summarized in Table 2.

From the results, Argos-Translate achieved the highest scores across all metrics, indicating its superior ability to generate accurate and contextually appropriate translations for our specific use case. In contrast, Gemini 1.5 flash showed relatively lower

**Table 2 Performance of translation models.**

| Translation model | BLEU | chrF | BERTScore F1 | METEOR |
|---|---|---|---|---|
| Google translate API | 0.0873 | 36.79 | 0.8138 | 0.2538 |
| Argos-Translate | 0.1168 | 38.45 | **0.8277** | 0.2694 |
| Gemini 1.5 Flash | 0.0178 | 22.58 | 0.7683 | 0.2157 |
| SILMA-9B-Instruct-v1.0 | 0.0728 | 33.15 | 0.8036 | 0.2131 |
| mBART-Large-50-MMT | 0.0832 | 35.54 | 0.8127 | 0.2473 |
| Marefa-MT-En-Ar | 0.0681 | 31.26 | 0.7787 | 0.1721 |

**Note:**
The best results are shown in bold.

scores, suggesting limitations in handling the required linguistic and contextual nuances for this task.

These findings highlight the importance of selecting suitable translation model based on their performance metrics to enhance the effectiveness of the Arabic plagiarism detection process.

## Plagiarism detection results

### Experiment configurations

To rigorously evaluate the effectiveness of the proposed plagiarism detection system, we conducted four distinct experiments. These experiments aimed to investigate the impact of two transformer-based architectures—AraT5 (*Elmadany, Nagoudi & Abdul-Mageed, 2023*) and Longformer (*Beltagy, Peters & Cohan, 2020*)—combined with two loss functions: Dice loss and weighted cross-entropy loss. Each experiment was performed with a defined context size, set at 1,024 tokens for AraT5 and 4,096 tokens for Longformer.

The configurations for the four experiments are as follows:

- **Experiment 1:** AraT5 with Dice loss and a context size of 1,024.
- **Experiment 2:** AraT5 with weighted cross entropy loss and a context size of 1,024.
- **Experiment 3:** Longformer with Dice loss and a context size of 4,096.
- **Experiment 4:** Longformer with weighted cross entropy loss and a context size of 4,096.

### Performance evaluation

The experimental results are summarized in Table 3. The table includes key findings from our experiments and comparisons with related state-of-the-art models.

The results reveal several important insights. Firstly, the AraT5 architecture consistently outperformed the Longformer in all experiments, regardless of the loss function applied. AraT5 achieved its best performance with the weighted cross-entropy loss, recording an F1-score of 0.9058, while Longformer reached a maximum F1-score of only 0.7563 under the same loss function. These observations suggest that the Arabic-specific pretraining of AraT5 offers a significant advantage in understanding the nuanced syntactic and semantic structures of Arabic texts.

**Table 3 Performance metrics for different configurations.**

| Model configuration | Precision | Recall | F1-score |
|---|---|---|---|
| AraT5 with dice loss | 0.896 | 0.912 | 0.899 |
| AraT5 with weighted cross entropy loss | **0.904** | **0.9217** | **0.9058** |
| Longformer with dice loss | 0.7512 | 0.7157 | 0.7214 |
| Longformer with weighted cross entropy loss | 0.7600 | 0.7596 | 0.7563 |
| Magooda_2 (*Bensalem et al., 2015*) | 0.852 | 0.831 | 0.8414 |
| Palkovskii_1 (*Bensalem et al., 2015*) | **0.977** | 0.542 | 0.6972 |
| Yes_semantic (*Zouaoui & Rezeg, 2022*) | 0.824 | 0.932 | **0.875** |
| No_semantic (*Zouaoui & Rezeg, 2022*) | 0.654 | **0.956** | 0.776 |

**Note:**
The best results are shown in bold.

Secondly, the choice of the loss function played a critical role in the system's performance. Both architectures demonstrated improved outcomes when trained using the weighted cross-entropy loss compared to the Dice loss. For AraT5, the transition from Dice loss to weighted cross-entropy loss resulted in an improvement from 0.899 to 0.9058 in F1-score. Similarly, Longformer benefited from a modest increase in F1-score, rising from 0.7214 to 0.7563. This improvement underscores the effectiveness of Weighted Cross-Entropy Loss in mitigating the impact of class imbalance, especially in datasets where certain plagiarism types are underrepresented.

Interestingly, despite the Longformer architecture's ability to handle significantly larger context windows (up to 4,096 tokens), it did not surpass the performance of AraT5, which was limited to a maximum context size of 1,024 tokens. This finding emphasizes that context size alone does not guarantee superior performance; instead, factors such as language-specific pretraining, model architecture, and fine-tuning strategies play a more pivotal role in achieving high accuracy.

When compared with state-of-the-art baselines, the proposed AraT5 with weighted cross-entropy loss achieved the highest F1-score among the evaluated models. For instance, it surpassed the Magooda_2 model (F1-score: 0.8414) and Yes_semantic (F1-score: 0.875). Although the Palkovskii_1 model achieved a remarkably high precision score (0.977), its recall was significantly lower (0.542), resulting in a poor overall F1-score (0.6972). These comparisons highlight the balanced nature of our approach, which maintains strong performance across precision, recall, and F1-score without overfitting to any single metric.

To ensure fair and consistent comparison with prior work, we focused on macro-averaged evaluation metrics—precision, recall, and F1-score—as these are the most commonly reported measures in previous Arabic plagiarism detection studies. While these metrics provide a robust overview of classification performance, we acknowledge that additional diagnostics such as confusion matrices and ROC-AUC curves could offer deeper insights into model behavior, including class-specific misclassifications and bias. We plan to include these in future iterations of this work to further strengthen the analysis.

In conclusion, the experimental results demonstrate that our proposed architecture, particularly AraT5 with weighted cross-entropy loss, offers a robust and reliable framework for plagiarism detection in Arabic text. It effectively addresses the challenges posed by dataset imbalance and contextual limitations while outperforming existing state-of-the-art solutions.

These findings not only validate the robustness of our approach but also provide insights for future research directions aimed at improving NLP systems tailored for underrepresented languages such as Arabic.

## CONCLUSION

This study presented a comprehensive approach to plagiarism detection tailored for Arabic and cross-lingual English-to-Arabic text, leveraging state-of-the-art transformer architectures and Siamese neural networks. By addressing key challenges such as dataset imbalances, context size limitations, and the semantic complexity of Arabic text, our approach introduces a structured pipeline that integrates preprocessing, advanced translation layers, and robust model architectures. The experimental evaluation demonstrated consistent performance improvements, underscoring the importance of architecture selection, fine-tuning, and appropriate loss functions in achieving reliable plagiarism detection outcomes.

### Strengths

The proposed system integrates Arabic-specific transformer models alongside scalable architectures capable of handling long text sequences, combining the linguistic strengths of AraT5 and the extended contextual abilities of Longformer. The incorporation of Siamese neural networks allowed for effective pairwise document comparison, while weighted cross-entropy and Dice loss addressed dataset imbalance challenges, enhancing model reliability. Furthermore, the translation layer bridges the gap for cross-lingual plagiarism detection, ensuring a broader applicability of the system across multilingual contexts.

### Limitations

Despite the promising results, the system faced limitations in both data availability and computational efficiency. The ExAraCorpusPAN2015 dataset, while valuable, remains limited in both size and label diversity, posing challenges in generalizing findings across broader domains. Additionally, the reliance on Longformer, which was not pre-trained specifically on Arabic data, introduced inherent limitations in capturing Arabic linguistic subtleties. Fine-tuning helped mitigate this to an extent, but an Arabic-pretrained Longformer remains a future need. Moreover, computational resource constraints restricted exploration into more extensive model ensembles or training deeper configurations.

### Opportunities for future research

Future directions can focus on developing larger, domain-diverse Arabic plagiarism datasets with balanced class distributions to overcome dataset limitations. Training an Arabic-specific Longformer model from scratch could address the contextual nuances of

long Arabic text sequences more effectively. Additionally, integrating hybrid architectures that combine transformer-based semantic analysis with statistical and linguistic features might further enhance performance. Exploring advanced multi-modal techniques, where textual data is complemented with metadata (*e.g.*, author style, citation patterns), could open new avenues for detecting more sophisticated forms of plagiarism.

In summary, this study establishes a robust foundation for Arabic plagiarism detection, addressing critical technical and methodological challenges, while laying the groundwork for future advancements in the field.

### Funding
The authors received no funding for this work.

### Competing Interests
The authors declare that they have no competing interests.

### Author Contributions
- Ahmad Abdelaal conceived and designed the experiments, performed the experiments, analyzed the data, performed the computation work, prepared figures and/or tables, authored or reviewed drafts of the article, and approved the final draft.
- Abdallah Elsaadany conceived and designed the experiments, performed the experiments, analyzed the data, performed the computation work, prepared figures and/or tables, authored or reviewed drafts of the article, and approved the final draft.
- Abdelrhman Ahmed Medhat conceived and designed the experiments, performed the experiments, analyzed the data, performed the computation work, prepared figures and/or tables, authored or reviewed drafts of the article, and approved the final draft.
- Aysha Al Shamsi analyzed the data, prepared figures and/or tables, and approved the final draft.
- Noha Gamal ElDin Saad Ali conceived and designed the experiments, analyzed the data, prepared figures and/or tables, authored or reviewed drafts of the article, and approved the final draft.

### Data Availability
The Arabic External Plagiarism Detection dataset is available at GitHub:
- https://github.com/ahmadsameh8/Plagiarism-Detection.
- Abdelaal, A., ElSaadany, A., Medhat, A., Al Shamsi, A., & Saad, N. G. E.-D. (2025). Plagiarism detection across languages: a comprehensive study of Arabic and English-to-Arabic long documents. Zenodo. https://doi.org/10.5281/zenodo.15733557.

The PAN Arabic External Plagiarism Detection Shared Task Corpus is available at Zenodo:

Bensalem, I., Boukhalfa, I., Rosso, P., & Chikhi, S. (2015). PAN Arabic External Plagiarism Detection Shared Task Corpus [Data set]. Arabic Plagiarism Detection Shared

Task (PAN@FIRE 2015) (AraPlagDet 2015). Zenodo. https://doi.org/10.5281/zenodo.6607799.

## Supplemental Information

Supplemental information for this article can be found online at http://dx.doi.org/10.7717/peerj-cs.3128#supplemental-information.

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
