# Peer review of "Plagiarism detection across languages: a comprehensive study of Arabic and English-to-Arabic long documents"

_PeerJ Computer Science, doi:10.7717/peerj-cs.3128_

## Round 0.1 · original submission · Major Revisions

·

Basic reporting

The authors of this paper propose an approach to Arabic plagiarism detection in a cross-lingual context, considering the English-Arabic language pair. The paper is well written and easy to read. The objective of the work is well stated, as well as the intended contributions of the authors. However, I have some remarks regarding the literature review:
- Regarding other Arabic NLP topics, there are few works in the topic of plagiarism detection, hence it is important to cite all relevant works. This work uses Siamese architecture, I think it is relevant to cite the works (which are very few) that first used this architecture. In the monolingual Arabic plagiarism detection section, the authors may add (they may be cited in the section siamese if not in monolingual PD as an illustration of using siamese for PD):
1. A. A. Aliane& H. Aliane (2020), "Evaluating SIAMESE Architecture Neural Models for Arabic Textual Similarity and Plagiarism Detection" .https://ieeexplore.ieee.org/abstract/document/9416550.
2. A. Mahmood & M. Zrigui (2022), "Siamese AraBERT-LSTM Model based Approach for Arabic Paraphrase Detection". https://aclanthology.org/2022.paclic-1.60.pdf
3. R. Saidi, F. Jarray, M. AlSuhaibani (2023) "SiameseBERT: A Bert-Based Siamese Network Enhanced with a Soft Attention Mechanism for Arabic Semantic Textual Similarity". https://www.semanticscholar.org/paper/d00b1fa1ddfdc902821d2d2f62fc0c1e3deeb7f7

- I think also that the section about cross-lingual plagiarism detection may be enhanced by explicating the process of this task before citing the related works. I think the work by Cherroun, H., Alshehri, A., et al. (2018) addresses disguised plagiarism in a monolingual context, hence, it is not relevant to cite it in this section. Some interesting works in cross-lingual English-Arabic plagiarism detection are missing in this section, which are available on Google Scholar. I think for such an understudied task, all existing references (they are really few) are relevant.


Experimental design

The authors describe a classical, correct experimental process to implement and evaluate their ideas, in particular regarding preprocessing and evaluation, which are well-known steps for any NLP task, but I have two main remarks here:

-. As mentioned by the authors, they use in this work the ExAraCorpusPAN2015, which corpus as mentioned by its authors, contains only documents in the Arabic language. Hence, what about the English data used in this work? Nothing is said about it. The authors may also describe the English data and give, for instance, statistics about the number of documents considered for translation. Only the studied translation tools and their evaluation results are given. As the paper is about English-Arabic cross-lingual PD, it is important to describe the English data to have an objective idea of both the translation layer result and the plagiarism detection evaluation as well.
- I understand that once the English documents are translated into Arabic, it becomes a monolingual PD, but it should be more relevant to compare your results with works addressing cross-lingual PD, especially, I think, they also implement a translation layer. You compared two monolingual PD works.
If you consider things at the monolingual stage (after translation), it would be more relevant to compare with the works that used Siamese (the first used transformers: AraBERT) or works that at least use recent neural models rather than traditional methods.

Validity of the findings

I cannot objectively give any judgment here because there are lacking elements:
- The English data (as I said above, the authors don't describe any English data; they just describe the tools they studied and their comparison)
- I don't find the comparison with other models relevant. I think it is important to compare with works addressing the cross-lingual context and with works that have used transformers or at least recent neural models.

Reviewer 2 ·

Basic reporting

The "Plagiarism Detection Across Languages: 2 A Comprehensive Study of Arabic and 3 English-to-Arabic Long Documents" document accurately and succinctly outlines the research objectives, methodology, and findings. It articulates the core problem—detection of plagiarism in Arabic texts—and proposes a solution using a combination of Siamese Neural Networks and transformer models (AraT5 and Longformer). The use of specific loss functions (Weighted Cross-Entropy and Dice Loss) to address class imbalance is noted accordingly.

However, the authors could have either employed a more recent corpus or provided a strong justification for retaining the earlier ExAraCorpusPAN2015. Although widely employed in earlier studies, since then, the research has expanded, and there are more recent corpora that are more diverse and extensive and capture modern writing styles as well as current detection issues, such as the 2022 ExAra Corpus or JUPlag Corpus. Not utilizing or acknowledging these newer datasets may limit the generalizability and timeliness of the study. Where some constraints led to the use of an older dataset, such constraints should have been properly stated by the authors to maximize the study's methodological clarity.

Experimental design

It is not certain how Figure 1 and Figure 3 have been derived from the same data set, since there appear to be differences in the label distribution and class representation shown in each. Figure 1 suggests a three-label distribution, while Figure 3 shows four distinct classes, leaving one wondering about the preprocessing steps or label transformation utilized between the two. The paper fails to adequately explain if another class was introduced during data preparation or if relabeling or a mix of original categories occurred. The validity and reliability of the experiment become difficult to quantify without clear reasoning or explanation behind this alteration. The authors must also explain the relationship between the numbers and provide an in-depth account of how the dataset evolved throughout the preprocessing pipeline.

Validity of the findings

The paper explains precision, recall, and F1 score, which are appropriate for quantifying classification performance, particularly when working with imbalanced datasets. But it omits critical diagnostic measures such as the confusion matrix and AUC-ROC curves, which are common in classification problems and would better explain the behavior of the model. A confusion matrix would help describe how the model acts with regard to particular classes, more so with the transition from three to four labels, and AUC-ROC plots could give information on the discriminative power of the model across the range of thresholds. The inclusion of these plots would supplement the evaluation by highlighting any biases, misclassifications, or class-associated weaknesses that do not show themselves using aggregate scores.
The manuscript lacks any form of statistical analysis, which is a significant omission given the performance claims made. There is no indication of variability (e.g., standard deviation, confidence intervals) or statistical significance testing to support the comparison between model configurations. This is particularly important when comparing similar models or loss functions, such as AraT5 with Weighted Cross-Entropy Loss versus Dice Loss, as observed differences may not be meaningful without statistical validation. The absence of such analysis limits the reliability and generalizability of the reported findings. The authors should include statistical tests, such as paired t-tests or bootstrap resampling, and report variance across multiple runs to ensure their results are not due to random fluctuations or overfitting.

---

## Round 0.2 · accepted · Accept

Dear Authors,

It is to be noted that one of the preceding reviewers elected not to respond to the invitation to review the revised paper. It is evident that one of the reviewers has accepted the paper in its current form. Furthermore, it is believed that the quality of the paper has been adequately enhanced.

Best wishes,

Reviewer 2 ·

Basic reporting

The authors clearly explained the reasons for the use of this dataset.

Experimental design

The authors have changed the labels of the figures and clarified the issue.

Validity of the findings

I am looking forward to seeing the future work that will incorporate more detailed statistical analysis.